# Pulsed Electromagnetic Field Transmission through a Small Rectangular Aperture: A Solution Based on the Cagniard–DeHoop Method of Moments

**Martin Štumpf** 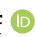

Lerch Group of EM Research, Department of Radioelectronics, FEEC, Brno University of Technology, Technická 3082/12, 616 00 Brno, Czech Republic; martin.stumpf@centrum.cz

**Abstract:** Pulsed electromagnetic (EM) field transmission through a relatively small rectangular aperture is analyzed with the aid of the Cagniard–deHoop method of moments (CdH-MoM). The classic EM scattering problem is formulated using the EM reciprocity theorem of the time-convolution type. The resulting TD reciprocity relation is then, under the assumption of piecewise-linear, space–time magnetic-current distribution over the aperture, cast analytically into the form of discrete time-convolution equations. The latter equations are subsequently solved via a stable marching-on-in-time scheme. Illustrative examples are presented and validated using a 3D numerical EM tool.

**Keywords:** computational electromagnetics; numerical analysis; electromagnetic transient scattering; time domain; Cagniard-DeHoop technique; Cagniard–deHoop method of moments





## 1. Introduction

Apertures in conducting planes are frequently encountered in the form of windows, cracks around doors, coupling slots in microwave devices, or imperfect seams between two metallic plates (e.g., ([1], Chapter 4) and ([2], Sections 6.7 and 7.9)). In order to quantify both intentional and undesired effects of such apertures, their EM scattering is a major concern in applied electromagnetics.

Wavefield penetration through an aperture in a thin screen is a classical problem of wavefield physics, solutions of which can be traced back to the work of Lord Rayleigh (see [3] and ([4], § 6.1)). His solution applying to the plane-wave diffraction by a relatively small (with respect to the wavelength) aperture was later extended by Bouwkamp and Van Bladel [5–7]. Without restricting himself to the plane-wave excitation, Bethe [8] demonstrated that the EM scattering by a small aperture can be, in certain circumstances (see ([5], § 9)), attributed to the action of equivalent electric and dipole moments [9,10]. Whenever the maximum linear dimension of the aperture is not sufficiently small with respect to the operating wavelength, however, the approximate aperture-diffraction models due to Rayleigh and Bethe are no longer applicable and one has to resort to more general formulations. A way out of the difficulty has been offered by Levine and Schwinger through their variational formulation [11,12], thus extending the steady-state solution to a wide range of frequencies.

The available rigorous solutions of the aperture diffraction problem were derived under the assumption of sinusoidally in time-varying wavefields. However, owing to the widespread use of communication and radar systems relying on the transmission, detection, and subsequent interpretation of digital signals, the steady-state assumption may no longer be computationally efficient and/or physically legitimate. Therefore, in the present work, we remove the restriction and solve the diffraction wavefield problem in its (original) space–time domain. The vast majority of previous works aiming at TD solutions rely more or less on the pertaining steady-state results—exact, purely TD analytical solutions of the aperture

diffraction are lacking in the literature on the subject. Indeed, the pertinent solutions can be, broadly speaking, divided into two categories. The first line of reasoning employs purely numerically approaches such as the inverse FFT algorithm [13,14] or the finite-difference TD technique [15,16]. The second approach yields TD approximate field expressions based on the use of Bethe's model [17] or Kirchhoff's approximation ([18], Section 5.5.1). To deal with the issue, the classic CdH technique [19] (see also [20–23], for example) and the TD EM reciprocity theorem of the time-convolution type ([24], Section 28.2) (see also ([25], Section 5.2), ([26], Section 1.4.1), and [27]) are in the present work combined to introduce a novel, rigorous TD integral-equation approach to analyzing the pulsed EM scattering by a relatively small rectangular aperture in a PEC screen. The new solution strategy, to be referred to as the CdH-MoM, has been previously applied to TD performance studies of cylindrical and planar antennas [28] and, more recently, to the TD EM scattering analysis of transmission lines in the presence of thin sheets [29] and EM metasurfaces [30,31].

## 2. Problem Definition

The problem configuration under analysis is shown in Figure 1. The position in this configuration is specified by the coordinates $\{x, y, z\}$ with respect to an orthogonal, Cartesian reference frame with the origin $\mathcal{O}$ and the three base vectors, $\{\boldsymbol{i}_x, \boldsymbol{i}_y, \boldsymbol{i}_z\}$, forming in the indicated order a right-handed triad. The position vector is then written as $\boldsymbol{r} = x\boldsymbol{i}_x + y\boldsymbol{i}_y + z\boldsymbol{i}_z$. The time coordinate is $t$, and the continuous time-convolution operator is represented by $*$. The Heaviside unit-step function is denoted by $H(t)$, and the Dirac-delta distribution is $\delta(t)$. Partial differentiation is denoted by $\partial$, which is supplied with the pertaining subscript. For instance, to differentiate with respect to $x$, we use $\partial_x$.

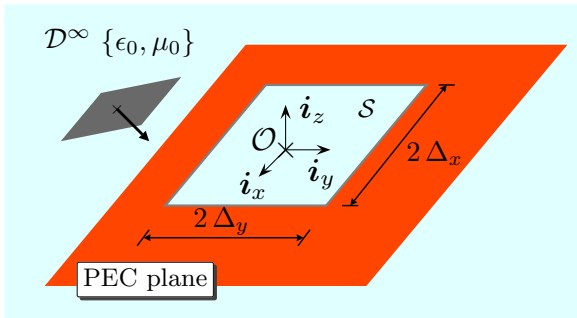

**Figure 1.** Rectangular aperture in a PEC plane.

We shall analyze the EM-field penetration through a bounded aperture in an un-bounded PEC screen. The PEC plane lies in $z = 0$, and the aperture occupies a rectangular domain $\mathcal{S} = \{-\Delta_x < x < \Delta_x, -\Delta_y < y < \Delta_y, z = 0\}$ with $\Delta_{x,y} > 0$. In the present analysis, it is assumed that the maximum dimension of the aperture is relatively small with respect to the spatial support of the exciting EM pulse. The surrounding medium is assumed to be linear, isotropic, and loss-free in its EM properties. It is described by (real-valued, positive scalar) electric permittivity $\epsilon_0$ and magnetic permeability $\mu_0$. The corresponding EM wave speed is $c_0 = (\epsilon_0 \mu_0)^{-1/2} > 0$, and the wave admittance is denoted by $Y_0 = (\epsilon_0 / \mu_0)^{1/2} > 0$.

The aperture is supposed to be irradiated by the (causal) incident EM wave field (denoted by superscript $^\mathrm{i}$), which is specified by its electric-field strength, $\boldsymbol{E}^\mathrm{i}(\boldsymbol{r}, t)$, and magnetic-field strength, $\boldsymbol{H}^\mathrm{i}(\boldsymbol{r}, t)$. The incident EM field is not necessarily a plane-wave. To account for the presence of the aperture, we next define the scattered EM wave field (denoted by superscript $^\mathrm{s}$) as the difference between the total fields, $\{\boldsymbol{E}, \boldsymbol{H}\}(\boldsymbol{r}, t)$, and excitation EM wave fields (denoted by superscript $^\mathrm{e}$), viz.

$$\{\boldsymbol{E}^\mathrm{s}, \boldsymbol{H}^\mathrm{s}\}(\boldsymbol{r}, t) = \{\boldsymbol{E} - \boldsymbol{E}^\mathrm{e}, \boldsymbol{H} - \boldsymbol{H}^\mathrm{e}\}(\boldsymbol{r}, t), \tag{1}$$

for all $\boldsymbol{r} \in \mathbb{R}^3$ and $t > 0$, where the excitation fields have the meaning of total fields in the *absence* of the aperture (i.e., with the "short-circuited aperture"). Since $\boldsymbol{i}_z \times \boldsymbol{E}^\mathrm{e}(\boldsymbol{r}, t) = \boldsymbol{0}$ over

the entire $z = 0$ plane for all $t > 0$, the use of the explicit-type boundary condition applying to the total EM field on the PEC screen in Equation (1) implies that $i_z \times E^s(r, t) = 0$ on the PEC screen for all $t > 0$.

## 3. Time Domain Problem Formulation

The EM scattering problem under consideration is formulated using the TD Lorentz reciprocity theorem ([24], Section 28.2). To that end, the reciprocity theorem is applied to the scattered EM wave fields and the (causal) testing EM wave fields (denoted by superscript $^T$), which are generated by the testing magnetic-current surface density, $\partial K^T(r, t)$, whose spatial support is the surface of the aperture, supp$(\partial K^T) = \mathcal{S}$. Then, upon applying the TD reciprocity theorem to the upper half-space, $z > 0$, while enforcing the surface boundary condition applying to (the tangential component of) the scattered field, $\partial K^s = E^s \times i_z$, on the PEC plane, we arrive at

$$\int_{r \in \mathcal{S}^+} \big[ H^T(x, y, 0^+, t) \overset{*}{\cdot} \partial K^s(x, y, t)$$

$$- H^s(x, y, 0^+, t) \overset{*}{\cdot} \partial K^T(x, y, t) \big] dA = 0, \tag{2}$$

where $\cdot$ represents the standard inner product of two vectorial quantities ([26], Equation (1.2)), $*$ denotes the continuous time-convolution operator ([26], Equation (1.11)), and the superscript $^+$ indicates that we approach the surface of the aperture, $\mathcal{S}$, from above as $z \downarrow 0$. Since both states are causal and each other's adjoint, the contribution from the bounding sphere "at infinity" vanishes ([26], Section 1.4.3). In the second step, the TD reciprocity theorem is applied to the lower half-space, $z < 0$, which leads to a similar relation:

$$\int_{r \in \mathcal{S}^-} \big[ H^T(x, y, 0^-, t) \overset{*}{\cdot} \partial K^s(x, y, t)$$

$$- H^s(x, y, 0^-, t) \overset{*}{\cdot} \partial K^T(x, y, t) \big] dA = 0, \tag{3}$$

where $^-$ indicates that we approach the surface of the aperture from below as $z \uparrow 0$. Recalling the definition of the excitation field, we may use Equation (1) with $i_z \times H^e(x, y, 0^+, t) = 2 i_z \times H^i(x, y, 0, t)$ and $i_z \times H^e(x, y, 0^-, t) = 0$ and with the continuity-type condition $i_z \times H(x, y, 0^+, t) = i_z \times H(x, y, 0^-, t)$ applying to all $r \in \mathcal{S}$ and $t > 0$, to combine relations (2) and (3). This yields the desired TD reciprocity relation:

$$\int_{r \in \mathcal{S}} H^T(x, y, 0^+, t) \overset{*}{\cdot} \partial K^s(x, y, t) dA$$

$$= -\int_{r \in \mathcal{S}} H^i(x, y, 0, t) \overset{*}{\cdot} \partial K^T(x, y, t) dA. \tag{4}$$

The final TD relation (4) will next be solved for the equivalent magnetic-current space–time distribution, $\partial K^s(x, y, t)$, as induced in the aperture. Owing to the relatively small dimension of the aperture, we may assume that the magnetic-current spatial distribution has the following form:

$$\partial K^s_x(x, y, t) = \frac{k_x(t)}{2\Delta_y} \Lambda(x) \Pi(y), \tag{5}$$

$$\partial K^s_y(x, y, t) = \frac{k_y(t)}{2\Delta_x} \Lambda(y) \Pi(x), \tag{6}$$

where $k_x(t)$ and $k_y(t)$ are the magnetic-current pulse shapes (to be computed),

$$\Lambda(x) = \begin{cases} 1 + x/\Delta_x, & \text{for } \{-\Delta_x \leq x \leq 0\} \\ 1 - x/\Delta_x, & \text{for } \{0 \leq x \leq \Delta_x\} \\ 0, & \text{elsewhere} \end{cases} \tag{7}$$

is the triangular function, and

$$\Pi(x) = \begin{cases} 1, & \text{for } \{-\Delta_x \leq x \leq \Delta_x\} \\ 0, & \text{elsewhere} \end{cases} \tag{8}$$

represents the rectangular function. The spatial distribution of the magnetic-current as represented by Equations (5) and (6) is chosen such that the pertinent end conditions, $\partial K_x^{\mathrm{s}} = 0$ and $\partial K_y^{\mathrm{s}} = 0$ at $x = \pm\Delta_x$ and $y = \pm\Delta_y$, respectively, are satisfied for all $t > 0$. Along their transverse direction, the magnetic-current components are assumed to have the uniform spatial distribution. Consequently, the magnetic-current flowing in parallel to the edge does not exhibit the inverse-square-root singularity as required by the pertaining edge condition ([1], Chapter 4). It can be expected, however, that this choice will have a negligible impact on the estimation of far-field characteristics [32]. Finally, it remains to specify the testing currents on the right-hand side of the TD reciprocity relation. To this end, we may choose the "razor-type" testing functions

$$\partial K_x^{\mathrm{T}}(x,y,t) = \Pi(2x)\delta(y)\delta(t), \tag{9}$$
$$\partial K_y^{\mathrm{T}}(x,y,t) = \Pi(2y)\delta(x)\delta(t). \tag{10}$$

It is noted that the thus-chosen testing currents satisfy the end conditions, $\partial K_x^{\mathrm{T}} = 0$ and $\partial K_y^{\mathrm{T}} = 0$ at $x = \pm\Delta_x$ and $y = \pm\Delta_y$, respectively.

## 4. Problem Solution

The problem will be solved via the CdH joint transform technique [19]. Accordingly, we combine a unilateral Laplace transformation, i.e.,

$$\hat{\boldsymbol{H}}(x,y,z,s) = \int_{t=0}^{\infty} \exp(-st)\boldsymbol{H}(x,y,z,t)\mathrm{d}t, \tag{11}$$

for $\{s \in \mathbb{R}; s > 0\}$, by virtue of Lerch's uniqueness theorem ([26], Appendix), with the wave slowness representation taken along the surface parallel to the PEC screen:

$$\hat{\boldsymbol{H}}(x,y,z,s) = \left(\frac{s}{2\pi\mathrm{i}}\right)^2 \int_{\kappa=-\mathrm{i}\infty}^{\mathrm{i}\infty} \mathrm{d}\kappa \int_{\sigma=-\mathrm{i}\infty}^{\mathrm{i}\infty} \exp[-s(\kappa x + \sigma y)]\tilde{\boldsymbol{H}}(\kappa,\sigma,z,s)\mathrm{d}\sigma, \tag{12}$$

where $\kappa$ and $\sigma$ are (imaginary-valued) slowness parameters in the $x$- and $y$-direction, respectively. As a matter of fact, Equation (12) is a two-dimensional Fourier inversion integral, where the (real-valued and positive) Laplace-transform parameter, $s$, plays the role of a scaling parameter. This representation entails the properties $\partial_x \to -s\kappa$ and $\partial_y \to -s\sigma$. Under the wave slowness representation, the TD reciprocity relation (4) can be written as

$$\begin{aligned} &\left(\frac{s}{2\pi\mathrm{i}}\right)^2 \int_{\kappa=-\mathrm{i}\infty}^{\mathrm{i}\infty} \mathrm{d}\kappa \int_{\sigma=-\mathrm{i}\infty}^{\mathrm{i}\infty} \left[\tilde{H}_x^{\mathrm{T}}(\kappa,\sigma,0^+,s)\partial\tilde{K}_x^{\mathrm{s}}(-\kappa,-\sigma,s)\right. \\ &\qquad\qquad\qquad\qquad\qquad \left.+ \tilde{H}_y^{\mathrm{T}}(\kappa,\sigma,0^+,s)\partial\tilde{K}_y^{\mathrm{s}}(-\kappa,-\sigma,s)\right]\mathrm{d}\sigma \\ &= -\left(\frac{s}{2\pi\mathrm{i}}\right)^2 \int_{\kappa=-\mathrm{i}\infty}^{\mathrm{i}\infty} \mathrm{d}\kappa \int_{\sigma=-\mathrm{i}\infty}^{\mathrm{i}\infty} \left[\tilde{H}_x^{\mathrm{i}}(\kappa,\sigma,0^+,s)\partial\tilde{K}_x^{\mathrm{T}}(-\kappa,-\sigma,s)\right. \\ &\qquad\qquad\qquad\qquad\qquad \left.+ \tilde{H}_y^{\mathrm{i}}(\kappa,\sigma,0^+,s)\partial\tilde{K}_y^{\mathrm{T}}(-\kappa,-\sigma,s)\right]\mathrm{d}\sigma. \end{aligned} \tag{13}$$

The (tangential components of the) transform domain testing fields as generated by the equivalent magnetic-current surface density, $\partial\tilde{K}^{\mathrm{T}}$, can be determined from the transform domain counterparts of standard source-type representations for the electric-

and magnetic-field strengths in a loss-free medium (cf. ([24], Equations (26.10–14) and (26.10–15))) as

$$\tilde{E}_x^{\mathrm{T}}(\kappa, \sigma, z, s) = \partial \tilde{K}_y^{\mathrm{T}}(\kappa, \sigma, s) \partial_z \tilde{G}(\kappa, \sigma, z, s), \tag{14}$$

$$\tilde{E}_y^{\mathrm{T}}(\kappa, \sigma, z, s) = -\partial \tilde{K}_x^{\mathrm{T}}(\kappa, \sigma, s) \partial_z \tilde{G}(\kappa, \sigma, z, s), \tag{15}$$

$$\tilde{H}_x^{\mathrm{T}}(\kappa, \sigma, z, s) = -(s/\mu_0)\Omega_0^2(\kappa)\partial \tilde{K}_x^{\mathrm{T}}(\kappa, \sigma, s)\tilde{G}(\kappa, \sigma, z, s)$$
$$+ (s/\mu_0)\kappa\sigma \partial \tilde{K}_y^{\mathrm{T}}(\kappa, \sigma, s)\tilde{G}(\kappa, \sigma, z, s), \tag{16}$$

$$\tilde{H}_y^{\mathrm{T}}(\kappa, \sigma, z, s) = -(s/\mu_0)\Omega_0^2(\sigma)\partial \tilde{K}_y^{\mathrm{T}}(\kappa, \sigma, s)\tilde{G}(\kappa, \sigma, z, s)$$
$$+ (s/\mu_0)\sigma\kappa \partial \tilde{K}_x^{\mathrm{T}}(\kappa, \sigma, s)\tilde{G}(\kappa, \sigma, z, s), \tag{17}$$

where $\Omega_0^2(\kappa) = 1/c_0^2 - \kappa^2$, and the transform domain Green's function, $\tilde{G}$, follows as the bounded solution of

$$\left(\partial_z^2 - s^2\gamma^2\right)\tilde{G} = 0 \text{ with } \lim_{z\downarrow 0}\partial_z\tilde{G} = -1. \tag{18}$$

Using the limit from Equation (18) in Equations (14) and (15), it can be easily verified that the transform domain testing fields satisfy the excitation conditions $\tilde{E}_x^{\mathrm{T}}(\kappa, \sigma, 0^+, s) = -\partial \tilde{K}_y^{\mathrm{T}}(\kappa, \sigma, s)$ and $\tilde{E}_y^{\mathrm{T}}(\kappa, \sigma, 0^+, s) = \partial \tilde{K}_x^{\mathrm{T}}(\kappa, \sigma, s)$. The (bounded) transform domain Green's function can be written as

$$\tilde{G}(\kappa, \sigma, z, s) = \exp[-s\gamma(\kappa, \sigma)z]/s\gamma(\kappa, \sigma), \tag{19}$$

for $z > 0$, where $\gamma = \gamma(\kappa, \sigma) = (1/c_0^2 - \kappa^2 - \sigma^2)^{1/2}$ with $\mathrm{Re}(\gamma) \geq 0$. Equation (19) can be subsequently used in the transform domain expressions (16) and (17) to determine the testing fields on the left-hand side of the reciprocity relation (13). The latter relation is subsequently solved in the TD. To that end, we first expand the unknown pulses $k_{x,y}(t)$ (see Equations (5) and (6)) in a piecewise linear manner:

$$k_x(t) = \sum_{k=1}^{M} v_{k;x}\Lambda_k(t), \tag{20}$$

$$k_y(t) = \sum_{k=1}^{M} v_{k;y}\Lambda_k(t), \tag{21}$$

where $v_{k;x}$ and $v_{k;y}$ are (yet unknown) coefficients and $\Lambda_k(t)$ represents the temporal triangular function:

$$\Lambda_k(t) = \begin{cases} 1 + (t - t_k)/\Delta_t, & \text{for } \{t_{k-1} \leq t \leq t_k\} \\ 1 - (t - t_k)/\Delta_t, & \text{for } \{t_k \leq t \leq t_{k+1}\} \\ 0, & \text{elsewhere} \end{cases} \tag{22}$$

along the discretized time axis $\{t_k = k\Delta_t; \Delta_t > 0, k = 1, 2, \cdots, M\}$. Finally, upon substituting the transform domain images of Equations (5) and (6) with (20) and (21), (9) and (10) in the reciprocity relation (13), we end up with a system of equations in the $s$ domain. Its constituent can be transformed to the original TD analytically via the CdH technique, which yields the following system of discrete time-convolution equations:

$$\sum_{k=1}^{m} \left(\underline{\mathcal{Y}}_{m-k+1} - 2\underline{\mathcal{Y}}_{m-k} + \underline{\mathcal{Y}}_{m-k-1}\right) \cdot V_k = I_m, \tag{23}$$

where $\underline{\mathcal{Y}}_k = \underline{\mathcal{Y}}(t_k)$ represents a $[2 \times 2]$ TD admittance array that is specified in Appendix A. Furthermore, $V_k = [v_{k;x}, v_{k;y}]^T$ is the $[2 \times 1]$ array of the unknown coefficients (see Equations (20) and (21)), and finally, $I_k = I(t_k) = [I_x(t_k), I_y(t_k)]^T$ denotes a

TD excitation $[2 \times 1]$ array, which, as a matter of fact, corresponds to the interaction term on the right-hand side of the TD reciprocity relation (4). Its specific form will be given in Section 5. Once the admittance and excitation arrays are specified, Equation (23) can be readily solved via the marching-on-in-time technique. This way leads to the following step-by-step updating scheme:

$$
V_m = \underline{\mathcal{Y}}_1^{-1} \cdot \left[ I_m - \sum_{k=1}^{m-1} \left( \underline{\mathcal{Y}}_{m-k+1} - 2\underline{\mathcal{Y}}_{m-k} + \underline{\mathcal{Y}}_{m-k-1} \right) \cdot V_k \right], \tag{24}
$$

which yields the desired coefficients for all $m = \{1, 2, \cdots, M\}$. Illustrative numerical examples validating the TD solution (24) are presented in the following section.

## 5. Illustrative Examples

Throughout this section, it is assumed that the rectangular aperture is irradiated by a uniform *E*-polarized TD plane-wave:

$$
\boldsymbol{E}^{\mathrm{i}}(\boldsymbol{r}, t) = \boldsymbol{\alpha} e^{\mathrm{i}}(t - \kappa_0 x - \sigma_0 y + \gamma_0 z), \tag{25}
$$

where the unit vector in the direction of polarization, $\boldsymbol{\alpha}$, can be expressed via the azimuthal angle, $\phi$, as $\boldsymbol{\alpha} = \sin(\phi)\boldsymbol{i}_x - \cos(\phi)\boldsymbol{i}_y$. The pertaining slowness parameters are then described by $\kappa_0 = \cos(\phi)\sin(\theta)/c_0$, $\sigma_0 = \sin(\phi)\sin(\theta)/c_0$ and $\gamma_0 = \cos(\theta)/c_0$, where $\theta$ denotes the polar angle. The (causal) plane-wave signature, $e^{\mathrm{i}}(t)$, has the property $e^{\mathrm{i}}(t) = 0$ for all $t < 0$. With reference to the right-hand side of the TD reciprocity relation (4), the corresponding tangential components of the magnetic-field strength at the plane of aperture, $z = 0$, then follow as

$$
H_x^{\mathrm{i}}(x, y, 0, t) = -Y_0 e^{\mathrm{i}}(t - \kappa_0 x - \sigma_0 y) \cos(\phi) \cos(\theta), \tag{26}
$$

$$
H_y^{\mathrm{i}}(x, y, 0, t) = -Y_0 e^{\mathrm{i}}(t - \kappa_0 x - \sigma_0 y) \sin(\phi) \cos(\theta). \tag{27}
$$

To determine the corresponding TD excitation array $\boldsymbol{I}(t)$ (see Equation (23)), we may use either Equations (9) and (10) with (26) and (27) on the right-hand side of Equation (4) or, equivalently, evaluate the right-hand side of the transform domain reciprocity relation (13). Both ways yield the following elements of the excitation array:

$$
I_x(t) = \frac{Y_0}{\kappa_0} \left[ \int_{\tau=0}^{t+\kappa_0 \Delta_x/2} e^{\mathrm{i}}(\tau) \mathrm{d}\tau - \int_{\tau=0}^{t-\kappa_0 \Delta_x/2} e^{\mathrm{i}}(\tau) \mathrm{d}\tau \right] \cos(\phi) \cos(\theta), \tag{28}
$$

$$
I_y(t) = \frac{Y_0}{\sigma_0} \left[ \int_{\tau=0}^{t+\sigma_0 \Delta_y/2} e^{\mathrm{i}}(\tau) \mathrm{d}\tau - \int_{\tau=0}^{t-\sigma_0 \Delta_y/2} e^{\mathrm{i}}(\tau) \mathrm{d}\tau \right] \sin(\phi) \cos(\theta). \tag{29}
$$

The limits pertaining to the normal incidence characterized by $\theta = 0$, implying $\kappa_0 = \sigma_0 = 0$, follow as

$$
\lim_{\theta \to 0} I_x(t) = Y_0 \Delta_x e^{\mathrm{i}}(t) \cos(\phi), \tag{30}
$$

$$
\lim_{\theta \to 0} I_y(t) = Y_0 \Delta_y e^{\mathrm{i}}(t) \sin(\phi). \tag{31}
$$

For the presented example, we take $2\Delta_x = 0.10$ m and $\Delta_y = \Delta_x/2$ with $\theta = 0$ and $\phi = \pi/4$. Furthermore, the plane-wave signature has the shape of a bipolar triangle, which can be described by

$$
\begin{aligned}
e^{\mathrm{i}}(t) = (2e_{\mathrm{m}}/t_{\mathrm{w}}) \big[ &t\,\mathrm{H}(t) - 2(t - t_{\mathrm{w}}/2)\mathrm{H}(t - t_{\mathrm{w}}/2) \\
&+ 2(t - 3t_{\mathrm{w}}/2)\mathrm{H}(t - 3t_{\mathrm{w}}/2) - (t - 2t_{\mathrm{w}})\,\mathrm{H}(t - 2t_{\mathrm{w}}) \big],
\end{aligned} \tag{32}
$$

where we take $e_{\mathrm{m}} = 1.0\,\mathrm{kV/m}$ and $c_0 t_{\mathrm{w}} = 1.0\,\mathrm{m}$ (see Figure 2). Consequently, $c_0 t_{\mathrm{w}} = 10\max(2\Delta_x, 2\Delta_y)$, which implies that the aperture is relatively small, as assumed in our analysis.

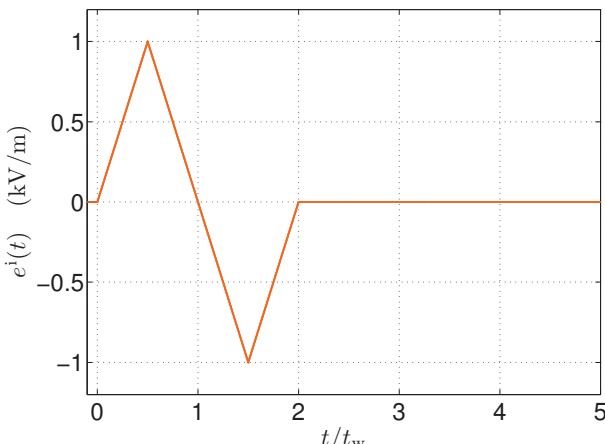

**Figure 2.** Incident plane-wave pulse shape.

In the first step, the marching-on-in-time solution (24) is applied to calculate the magnetic-current coefficients, $v_{k;x}$ and $v_{k;y}$, which, through Equations (20) and (21) with (5) and (6), determine the space–time distribution of the magnetic-current surface density induced throughout the aperture. Its tangential components can be expressed using the actual electric-field strengths in the aperture as $\partial K_x^{\mathrm{s}}(x,y,t) = E_y(x,y,0,t)$ and $\partial K_y^{\mathrm{s}}(x,y,t) = -E_x(x,y,0,t)$. The electric-field pulses at the center of the aperture as calculated with the aid of the CdH-MoM method and via the finite-integration technique (FIT) as implemented in CST Microwave Studio® are shown in Figure 3a,b. As can be seen, the results agree very well. To quantify the deviation between the signals, we evaluated the normalized root-mean-squared error according to ERR $= \sqrt{\sum_{k=1}^{M}(\bar{f}_k - f_k)^2/M}/\Delta f$, where $\bar{f}_k$ and $f_k$ represent the time samples of the FIT and CdH-MoM signals, respectively, and $\Delta f = f_{\max} - f_{\min}$. The error corresponding to the $E_x$-field component (see Figure 3a) is about ERR $\simeq 1.5\%$, while ERR $\simeq 3.2\%$ for the $E_y$-field component (see Figure 3b). The higher error in the $E_y$-field component can be attributed to the postulated magnetic-current space–time distribution (see Equations (5) and (6)), which is less accurate along the relatively longer side of the aperture.

Typical computational times to obtain the TD responses shown in Figure 3 are about 20 s using a non-optimized MATLAB® CdH-MoM implementation and about 30 min using CST Microwave Studio®. An even more striking difference is in the number of required unknowns and accompanying memory requirements. While our dedicated CdH-MoM computational model solves the system for two unknown quantities only (see Equations (5) and (6)), the FIT model consists of about 600 thousand discretization elements. The calculations were conducted on a standard laptop with Intel(R) Core(TM) i7-10510U CPU @ 1.80 GHz.

To indicate the range of applicability of the presented computational model, we increased the aperture's length along the $x$-direction such that $2\Delta_x$ is now a fifth of the excitation pulse's spatial support $c_0 t_{\mathrm{w}}$, i.e., $2\Delta_x = 0.20\,\mathrm{m} = c_0 t_{\mathrm{w}}/5$. In addition, we took $\Delta_y = \Delta_x/10$, so that the model represents a relatively narrow slit in the PEC screen. The remaining parameters were kept the same. Figure 4 shows the corresponding pulse shapes of the $E_y$-field at the center of the aperture. As the length of the slit is no longer sufficiently small with respect to $c_0 t_{\mathrm{w}}$, discrepancies between the signals, quantified by ERR $\simeq 5.8\%$ are apparent. While this result can still be useful for initial estimates, improving the accuracy through the incorporation of the piecewise-linear spatial expansion of the axial current density (e.g., ([28], Section 14.3)) is advisable in this case.

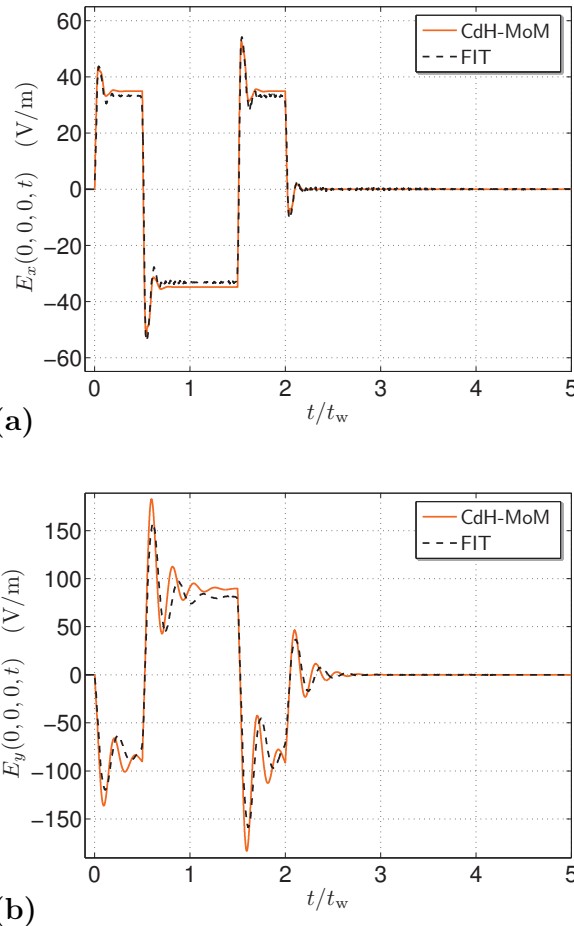

(a)

(b)

**Figure 3.** Electric-field pulse shapes as induced at the center of the aperture with relative dimensions $2\Delta_x/c_0 t_w = 1/10$ and $2\Delta_y/c_0 t_w = 1/20$. (**a**) $E_x$-field component; (**b**) $E_y$-field component.

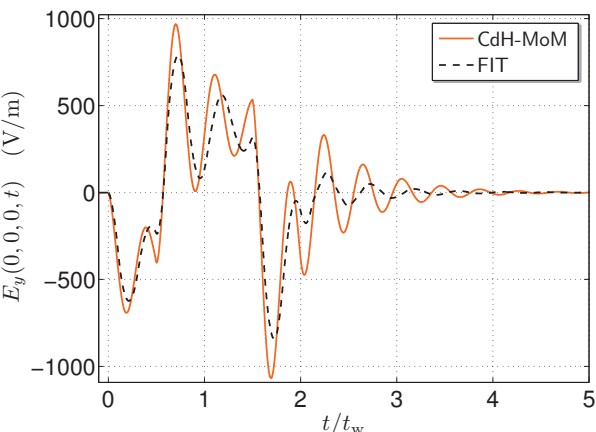

**Figure 4.** Pulse shape of $E_y$-field as induced at the center of the narrow aperture with relative dimensions $2\Delta_x/c_0 t_w = 1/5$ and $2\Delta_y/c_0 t_w = 1/50$.

Once the space–time distribution of the equivalent magnetic-current surface density in the aperture is known, it can be used to evaluate the scattered and, hence, total EM-fields via Equation (1). To that end, one may employ the pertaining EM wave field representations (e.g., ([24], Section 28.12)). For example, the TD scattered far-field amplitudes, $\{E^{s;\infty}, H^{s;\infty}\}$, defined via ([24], Section 26.12)

$$\{E^{\mathrm{s}}, H^{\mathrm{s}}\}(\boldsymbol{r}, t) = \frac{\{E^{\mathrm{s};\infty}, H^{\mathrm{s};\infty}\}(\boldsymbol{\xi}, t - |\boldsymbol{r}|/c_0)}{4\pi|\boldsymbol{r}|}\left[1 + O(|\boldsymbol{r}|^{-1})\right], \tag{33}$$

as $|\boldsymbol{r}| \to \infty$, with $\boldsymbol{\xi} = \boldsymbol{r}/|\boldsymbol{r}| = \cos(\phi^{\mathrm{s}})\sin(\theta^{\mathrm{s}})\boldsymbol{i}_x + \sin(\phi^{\mathrm{s}})\sin(\theta^{\mathrm{s}})\boldsymbol{i}_y + \cos(\theta^{\mathrm{s}})\boldsymbol{i}_z$ being the unit vector in the direction of observation, can be approximately evaluated from

$$E_x^{\mathrm{s};\infty}(\boldsymbol{\xi}, t) \simeq (2\Delta_y/c_0)\xi_z\partial_t k_y(t), \tag{34}$$

$$E_y^{\mathrm{s};\infty}(\boldsymbol{\xi}, t) \simeq -(2\Delta_x/c_0)\xi_z\partial_t k_x(t), \tag{35}$$

$$E_z^{\mathrm{s};\infty}(\boldsymbol{\xi}, t) \simeq (2\Delta_x/c_0)\xi_y\partial_t k_x(t) - (2\Delta_y/c_0)\xi_x\partial_t k_y(t), \tag{36}$$

where $k_x(t)$ and $k_y(t)$ directly result from the TD solution (24) via Equations (20) and (21). The corresponding magnetic-type far-field amplitudes simply follow from $H^{\mathrm{s};\infty}(\boldsymbol{\xi}, t) = Y_0\boldsymbol{\xi} \times E^{\mathrm{s};\infty}(\boldsymbol{\xi}, t)$. It is further worth noting that Equations (34)–(36) are, as a matter of fact, exact in the directions normal to the plane of aperture, $\theta^{\mathrm{s}} = 0$ and $\theta^{\mathrm{s}} = \pi$, and that the scattered wave fields in $z < 0$ are equal to the total fields transmitted through the aperture (cf. Equation (1)). For the sake of validation, the $z$-component of the far-field electric-type amplitude at $\phi^{\mathrm{s}} = \pi/4$ and $\theta^{\mathrm{s}} = 5\pi/6$ was evaluated via Equation (36) using the CdH-MoM solution (24) and with the help of a "far-field probe" in CST Microwave Studio®. Figure 5 shows the resulting pulse shapes. Again, the results correlate well with ERR $\simeq 6.2\%$.

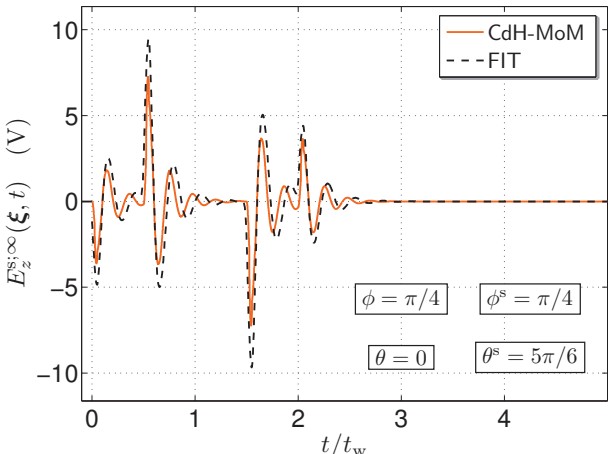

**Figure 5.** Electric far-field amplitude behind the aperture.

## 6. Conclusions

Using the EM reciprocity theorem of the time-convolution type and the CdH technique, we introduced a fundamentally new solution to the transient EM-field penetration through a relatively small aperture of a rectangular shape. It was demonstrated that for the postulated piecewise-linear space–time distribution of the equivalent magnetic-current surface density induced in the aperture, the pertaining TD admittance array can be derived analytically in terms of elementary functions only. The presented TD solution can be hence viewed as an exact "weak" solution of the EM aperture problem. Subsequently, the CdH-MoM solution was implemented and successfully verified using a commercial 3D EM computational tool. It turns out that the CdH-MoM solution is easy to implement and introduces huge computational savings of several orders of magnitude with respect to general-purpose, TD differential numerical approaches.

**Funding:** The research reported in this paper was financially supported by the Czech Science Foundation under Grant No. 20-01090S.

**Institutional Review Board Statement:** Not applicable.

**Informed Consent Statement:** Not applicable.

**Data Availability Statement:** The presented data are available upon request from the author.

**Conflicts of Interest:** The author declares no conflict of interest.

## Abbreviations

The following abbreviations are used in this manuscript:

| | |
|---|---|
| EM | Electromagnetic |
| CdH | Cagniard–deHoop |
| CdH-MoM | Cagniard–deHoop method of moments |
| PEC | Perfectly electrically conducting |
| TD | Time domain |

## Appendix A. Time Domain Admittance Array

Since the $x$- and $y$-components of the induced magnetic-current surface density are in a small rectangular aperture mutually uncoupled, the TD admittance array has only non-zero diagonal terms, viz.

$$\underline{\mathcal{Y}}(t) = \begin{bmatrix} Y_{xx}(t) & 0 \\ 0 & Y_{yy}(t) \end{bmatrix}. \tag{A1}$$

The elements of the admittance array can be for the piecewise-linear space–time distribution of the induced magnetic-current density (see Equations (5) and (6) with (20) and (21)) obtained analytically in the following form:

$$Y_{xx}(t) = \frac{Y_0}{c_0 \Delta_t \Delta_x \Delta_y} \Big[ \quad \Psi(3\Delta_x/2, \Delta_y, t) - \Psi(3\Delta_x/2, -\Delta_y, t) \\ -3\,\Psi(\Delta_x/2, \Delta_y, t) + 3\,\Psi(\Delta_x/2, -\Delta_y, t) \\ +3\,\Psi(-\Delta_x/2, \Delta_y, t) - 3\,\Psi(-\Delta_x/2, -\Delta_y, t) \\ -\Psi(-3\Delta_x/2, \Delta_y, t) + \Psi(-3\Delta_x/2, -\Delta_y, t) \Big], \tag{A2}$$

and $Y_{yy}(t)$ is given by a similar TD expression that can be obtained upon replacing $x$ with $y$ (and vice versa) in Equation (A2). The TD function, $\Psi(x, y, t)$, follows as the inverse of the complex-slowness integral (cf. ([28], (G.3))):

$$\hat{\Psi}(x, y, s) = \frac{c_0^2}{8\pi^2} \int_{\kappa \in \mathbb{K}_0} \frac{\exp(s\kappa x)}{s^3 \kappa^3} \Omega_0^2(\kappa) d\kappa \int_{\sigma \in \mathbb{S}_0} \frac{\exp(s\sigma y)}{s\sigma} \frac{d\sigma}{\gamma(\kappa, \sigma)} \tag{A3}$$

for $\{s \in \mathbb{R}; s > 0\}$, $\{x \in \mathbb{R}; x \neq 0\}$, $\{y \in \mathbb{R}; y \neq 0\}$, and recall that $\gamma(\kappa, \sigma) = (1/c_0^2 - \kappa^2 - \sigma^2)^{1/2}$. Furthermore, $\mathbb{K}_0$ and $\mathbb{S}_0$ are integration paths that run along the imaginary axes in the complex-slowness $\kappa$- and $\sigma$-planes, respectively, and that are indented to the right around their origins with small semi-circular arcs of vanishingly small radii (cf. ([28], Figure G.1)). The inversion of $\hat{\Psi}(x, y, s)$ can be carried out using the CdH procedure as closely described in ([28], Appendix G). First, the integrand in the integral with respect to $\sigma$ is continued analytically into the complex $\sigma$-plane, while keeping $\mathrm{Re}(\gamma) \geq 0$. This implies the horizontal branch cuts extending along $\{\mathrm{Im}(\sigma) = 0, \Omega_0(\kappa) < |\mathrm{Re}(\sigma)| < \infty\}$. Consequently, using Jordan's lemma and Cauchy's theorem [24] (p. 1054), path $\mathbb{S}_0$ is replaced with the loop around the branch cut, a parametric form of which is $\sigma(u) = -u\Omega_0(\kappa)\mathrm{sgn}(y) \pm i0$ for all $\{1 \leq u < \infty\}$ with $\mathrm{sgn}(y) = |y|/y$. In addition, the contribution from the simple pole at $\sigma = 0$ must be accounted for when $y \geq 0$. In the integral around the branch cut, we introduce the new variable of integration, $u$, while the integration around the pole is readily carried out analytically. The thus-transformed inner integral with respect to $\sigma$ is next substituted back in Equation (A3). Proceeding further in a similar manner with the integration in the complex $\kappa$-plane, we, after some lengthy, yet straightforward algebra, arrive at integral expressions that can be

evaluated analytically. Transforming finally the result of integration to the TD via standard tables ([33], Section 29), we end up with the TD original of Equation (A3) in the following form ([28], (cf. Equations (G.24), (G.25), and (G.30)):

$$
\begin{aligned}
\Psi(x,y,t) \;=\;& \frac{\mathrm{sgn}(x)\mathrm{sgn}(y)}{12\pi}\int_{v=r}^{c_0 t}(c_0 t - v)^3 f(x,y,v)\,\mathrm{d}v \\
&+ \frac{\mathrm{sgn}(y)\mathrm{H}(x)}{4\pi}\left\{ |y|\left(c_0^2 t^2 - x^2 + \frac{y^2}{3}\right)\cosh^{-1}\left(\frac{c_0 t}{|y|}\right) \right.\\
&\quad - c_0 t\left(\frac{c_0^2 t^2}{6} - x^2\right)\tan^{-1}\left[\left(\frac{c_0^2 t^2}{y^2} - 1\right)^{1/2}\right] \\
&\quad \left. - \frac{7}{6}c_0 t y^2\left(\frac{c_0^2 t^2}{y^2} - 1\right)^{1/2}\right\}\mathrm{H}(c_0 t - |y|) \\
&+ \frac{\mathrm{sgn}(x)\mathrm{H}(y)}{4\pi}\left\{ |x|\left(c_0^2 t^2 - \frac{x^2}{6}\right)\cosh^{-1}\left(\frac{c_0 t}{|x|}\right) \right.\\
&\quad - c_0 t\left(\frac{c_0^2 t^2}{6} - x^2\right)\tan^{-1}\left[\left(\frac{c_0^2 t^2}{x^2} - 1\right)^{1/2}\right] \\
&\quad \left. - \frac{5}{3}c_0 t x^2\left(\frac{c_0^2 t^2}{x^2} - 1\right)^{1/2}\right\}\mathrm{H}(c_0 t - |x|) \\
&+ \frac{c_0 t}{4}\left(\frac{c_0^2 t^2}{6} - x^2\right)\mathrm{H}(x)\mathrm{H}(y)\mathrm{H}(t),
\end{aligned}
\tag{A4}
$$

where we used $r = (x^2 + y^2)^{1/2} > 0$ with

$$
\begin{aligned}
f(x,y,v) \;=\;& \frac{1}{2v}\left[\frac{1}{(v^2/x^2-1)^{1/2}} + \frac{1}{(v^2/y^2-1)^{1/2}}\right] \\
&- \frac{1}{16}\frac{y^4}{x^2 v^3}\frac{1}{(v^2/y^2-1)^{5/2}}\left[3\frac{v^8}{y^8} + 6\frac{v^6}{y^6}\left(\frac{x^2}{y^2}-1\right)\right.\\
&\quad \left. + \frac{v^4}{y^4}\left(15\frac{x^4}{y^4} - 10\frac{x^2}{y^2} + 3\right) + 4\frac{v^2}{y^2}\frac{x^2}{y^2}\left(1 - 5\frac{x^2}{y^2}\right) + 8\frac{x^4}{y^4}\right] \\
&- \frac{1}{2}\frac{x^2}{v^3}\frac{1}{(v^2/x^2-1)^{1/2}} - \frac{3}{16}\frac{v}{x^2}\frac{1}{(v^2/y^2-1)^{5/2}}\left(3\frac{v^4}{y^4}\right.\\
&\quad \left. + 2\frac{v^2}{y^2}\left(\frac{x^2}{y^2}-3\right) + \frac{x^2}{y^2}\left(3\frac{x^2}{y^2}-2\right) + 3\right) \\
&+ \frac{3}{4}\frac{v}{x^2}\frac{1}{(v^2/y^2-1)^{3/2}}\left(\frac{v^2}{y^2} + \frac{x^2}{y^2} - 1\right).
\end{aligned}
\tag{A5}
$$

The calculation of the convolution integral in Equation (A4) is computationally the most exacting task. An efficient way to remedy the issue is the recursive-convolution technique ([28], Appendix H). Alternatively, one may apply integration by parts and write

$$
\begin{aligned}
\int_{v=r}^{c_0 t}(c_0 t - v)^3 f(x,y,v)\,\mathrm{d}v =\;& -(c_0 t - r)^3 \partial_v^{-1} f(x,y,r) \\
& - 3(c_0 t - r)^2 \partial_v^{-2} f(x,y,r) - 6\,(c_0 t - r)\partial_v^{-3} f(x,y,r) \\
& - 6\,\partial_v^{-4} f(x,y,r) + 6\,\partial_v^{-4} f(x,y,c_0 t),
\end{aligned}
\tag{A6}
$$

where $\partial_v^{-n}$ denotes the $n$-th integration with respect to $v$. Thanks to the relatively simple form of $f(x,y,v)$ (see Equation (A5)), the required integrals can be readily found analytically, thus enabling exact and fast computation of the TD admittance array elements via Equations (A1), (A2), and (A4).

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
