# Peer review of "Pulsed Electromagnetic Field Transmission through a Small Rectangular Aperture: A Solution Based on the Cagniard–DeHoop Method of Moments"

_algorithms, doi:10.3390/a15060216_

Round 1

Reviewer 1 Report

Present work studies the pulsed electromagnetic field transmission through a relatively small rectangular aperture using the Cagniard-deHoop method of moments. The presentation of the problem is detailed, and the results of the solution of an indicative case (at two points of observation) are presented and validated with CST (FIT). The work is interesting and the paper is overall well written, albeit a bit dense, with an expectation from the reader to be at least mildly accustomed with CdH-MoM.

Please find my comments below.

1.  All figures should be cited in the main text as Figure X.

2.  I am unfamiliar with the notation (2)-(4) for inner product combined with convolution. The equations should be provided in detail.

3.  Since validation is mentioned, some kind of metric and its results for the comparison (Figures 3,4) should be included.

4.  Author should discuss the reason Ex has a better fit than Ey at the center of the aperture.

5.  It would be interesting to discuss the limits of the method with respect to the dimensions of the aperture and tw.

6.  This is more of a proposal but from an EMC aspect it would be interesting to see the solutions in additional cases of apertures (e.g., a square aperture, a x-slit aperture with φ =0 and φ= π/2 for the wave) and relatively close to the aperture in order to investigate the field coupling inside a unit/device? By slit, I mean an aperture of dimensions ratio over 5 or even 10. Maybe some results and discussion can be introduced to the manuscript? The slit case could be of practical importance to the EMC community. 

Reviewer 2 Report

The paper is investigating EM transmission through a small rectangular aperture based on the Cagniard-De Hoop method. The manuscript is well-written, its mathematical rigor is higher than average, and numerical results are excellent. Hence, it is recommended for publication subject to the following remarks:

1.       In eqs. (5) and (6) the pulse shapes along the x and y axis are different (triangular vs. rectangular). What is the reason of non-uniformity?

2.       Similarly, why are “razor-type” testing functions used in (9) and (10)?

3.       Eqs. (11) and (12) need some clarification, although they are based on existing literature[18]: Apparently (11) is a straightforward Laplace transform, a fact that is not mentioned in the text. Also (12) looks like a strange version of a double inverse Laplace transform with awkward integration limits (zero real part) and wrong sign in the exponent (+ was expected). After downloading [18] it turned out that (12) is actually a Fourier transform, where the original real variable of integration was replaced by an imaginary one! Please clarify this complicated situation.

4.       What is the point of (14) and (15), since they are not used in the analysis?

5.       In the definition of σ(u) in the end of the Appendix, there is a strange symbol like a Greek β multiplied by 0. Is this a typo?

6.       At the end of the conclusions, the author claims that this method is computationally far less expensive that standard TD numerical approaches. I have no reason to doubt this claim, but there is no proof for this in the paper. How does this method compare with others in terms of computational cost? Can the author quantify the improvement, showing memory requirements and CPU time? In particular, how much better is this method with respect to FIT, used as reference in Figs. 3 and 4?

Round 2

Reviewer 1 Report

I thank the author for his response and the revised manuscript.

The author has addressed all my comments. 

-Ref [26] and [28] are the same. One of them should be removed.